

**A reduced-order model for dual state-parameter geostatistical inversion**
Yu-Li Wang[1], Tian-Chyi Jim Yeh[1], Jui-Pin Tsai[2,*]
1. Department of Hydrology and Atmospheric Sciences, University of Arizona,
Arizona, United States.
2. Department of Bioenvironmental Systems Engineering, National Taiwan
University, Taiwan.
* Corresponding Author
Assistant Professor Jui-Pin Tsai, jptsai@ntu.edu.tw





**0. Abstract**

To properly account the subsurface heterogeneity, geostatistical inverse models
usually permit enormous amount of spatial correlated parameters to interpret the
collected states. Several reduced-order techniques for the brick domain are
investigated to leverage the memory burden of parameter covariance. Their capability
to irregular domain is limited. Furthermore, due to the over fitting of states, the
estimated parameters usually diverge to unreasonable values. Although some
propriate tolerances can be used to eliminate this problem, they are presumed and
heavily rely on the personal judgement. To address these two issues, we present a
model reduction technique to the irregular domain by singular value decomposition
(SVD). Afterward, the state errors and parameters are sequentially updated to leverage
the over fitting. The computational advantages of the proposed reduced-order dual
state-parameter inverse algorithm are demonstrated through two numerical
experiments and one case study in a catchment scale field site. The investigations
suggest that the stability of convergence dramatically improves. The estimated
parameter values stabilize to reasonable order of magnitude. In addition, the memory
requirement significantly reduces while the resolution of estimate preserves. The
proposed method benefits multi-discipline scientific problems, especially useful and
convenient for assimilating different types of measurements.



## 1. Introduction

Groundwater is one of the necessary resources in many regions where the
amount of rainfall and the capacity of reservoir is limited. To provide enough fresh
water for the current and future uses in these areas, proper water resources
management and contaminated site remediation strategies are required, which relies
on the understanding of the site-specific spatial distribution of hydrological
parameters (e.g., hydraulic conductivity and specific storage) in the prefer scale.
Many covariance based geostatistical approaches have been widely employed for
aquifer characterization. Several previous studies suggested that the geostatistical
inversion is superior than many other subsurface inverse modeling because it
estimates the uncertainty and has ability to assimilate different type of observed data
sequentially (Vesselinov et al., 2001). However, as pointed out by Illman et al. (2015),
when the number of observations and unknown parameters are huge, the primary
drawbacks of geostatistical inversion are the computational and memory burdens.
Several ensemble approaches have been proposed to handle the memory and
large covariance matrices. For instance, Particle Filter or Sequential Monte Carlo
method (SMC, Field et al., 2016; Zhang et al., 2017), iterative Ensemble Kalman
Filter (EnKF, Schöniger et al., 2012; Ait-El-Fquih et al., 2016), iterative Ensemble
Smoother (ES, Zhang et al., 2018), Extended Kalman Filter (EKF, Yeh and Huang
2005; Leng and Yeh, 2003), and many other related methods construct the covariance
between the parameter and state variable from a set of ensemble member. Since a
bunch of realizations (usually several hundreds or thousands) are required to infer the
population covariance from the sample covariance, the algorithm may not be
computational affordable when the simulation time of single forward modeling is time
consuming.
On the other hand, Quasi-Linear Geostatistical Approach (QLGA, Kitanidis,



1995) and Successive Linear Estimator (SLE, Yeh et al., 1996) avoid generating a
large set of ensemble realizations. They construct the parameter covariance by some
prior knowledge of unknown parameter field (e.g., covariance function, variance, and
correlation length). Afterward, the covariance between the parameter and state
variable is estimated through the sensitivity of state variable with respect to parameter.
This approach requires significant amount of memory resource when the number of
unknown parameter and state variable are huge. Furthermore, evaluating the
sensitivity efficiently may be a difficult task for some scientific problems. As a result,
considerable efforts are devoted to improving the capability of the algorithm. For
instance, Sun and Yeh (1990) employed the adjoint approach to evaluate the
sensitivity. It reduces the cost of running forward model from the order of number of
unknown parameters to the number of state measurements. Saibaba and Kitanidis
(2012) incorporates the hierarchical matrices technique with a matrix-free Krylov
subspace approach to improve the computational efficiency. Liu et al. (2014) avoids
the direct solution of sensitivity matrix by the Krylov subspace method. Li et al. (2015)
and Zha et al. (2018) project the covariance matrix on the orthonormal basis and
evaluate the cross product of sensitivity and squared root covariance directly using
finite differencing approach. This method eliminates the sensitivity evaluation and
reduces the computational cost of running forward model to the order of number of
leading modes. Li et al. (2014) take the advantage of hierarchical nature of matrices to
accelerate the computation of dense matrix vector products and rewrite the Kalman
filtering equations into a computational efficient manner. Ghorbanidehno et al. (2015)
extend their approach to the general case of non-linear dynamic systems. Similarly,
Lin et al. (2016) reduces the computational complexity by projecting the parameters
to different hierarchies of Krylov subspace. Pagh (2013) use fast Fourier transform to
speed up the computation of covariance matrix multiplication. In addition, many





approaches reduce the computational cost and memory requirement. For example,
Nowak and Litvinenko (2013) combine low rank approximations to the covariance
matrices with fast Fourier transform; Kitanidis (2015) decomposes the covariance
matrix by some orthonormal basis and shows that the choice of basis can be tailored
to the problem of interest to improve estimation accuracy; Li et al. (2015) use discrete
cosine transform to compress the data covariance matrix of a 1-D state variable series;
Zha et al. (2018) use Karhunen-Loeve Expansion to compress the parameter
covariance matrix of a 3-D parameter field. Other useful reduced order models are
Galerkin projection (Liu et al., 2013), principal component (Kitanidis and Lee, 2014),
randomized algorithm (Lin et al., 2017), Whittaker-Shannon interpolation (Horning et
al., 2019), and Kronecker product decomposition (Zunino and Mosegaard, 2019).

In addition to reformulate the covariance matrix, the temporal moments

eliminate the temporal derivative term in the governing equation. Thus, it is another
potential method to reduce the data size and computational cost (Cirpka and Kitanidis,
2000; Nowak and Cirpka, 2006; Yin and Illman, 2009).

There are several limitations exist in the previous geostatistical inverse

algorithms. The first issue is over calibration or over fitting. During the inverse
process, the calibration terminates when the difference between the observed and
simulated states reduces to the value smaller than the given tolerance, an arbitrary
value based on user's personal judgement. In practical, the tolerance is determined by
the expected numeric and measurement errors. Since its true order of magnitude is
unknow, the estimated parameter field sometimes diverges if the tolerance is
underestimated. To be specific, the estimated parameters will first converge to the best
values accompanied with the successive assimilation of the information about the
subsurface heterogeneity embedded in the observed state variables. The values of
parameter then diverge to the unreasonable huge or small values to compensate the





numeric and measurement errors. This instability is not user friendly because the
reasonable (i.e., converged) estimate needs to be selected manually. Furthermore,
when different types of measurement (e.g., water level, flux, temperature, gravity, etc.)
are available, it is suggested that assimilate these data sequentially is a more robust
approach than the simultaneous assimilation (Tsai et al., 2017). Accordingly, the
manually    determination    of    convergence    prohibits    the    automatic    sequential
assimilation.
Second, when dealing with a 2-D or 3-D parameter or state variable fields, a
specific matrix structures are required to efficiently decompose the unconditional
covariance matrix to the orthonormal basis. For instance, a regular grid spacing is
required to efficiently perform the fast Fourier transform (Nowak and Litvinenko,
2013) and discrete cosine transform (Li et al., 2015). Similarly, Karhunen-Loeve
Expansion (Zha et al., 2018) requires a brick or rectangle shape domain and grid. This
requirement comes from the derivation of analytic eigenvalue and eigenvector of a
separable exponential function.
To overcome these two existing limitations, we first introduce an additional step
to estimate the error of state variables based on the error covariance matrices. Next,
we derive a reduced order model using singular value decomposition. Afterward, we
present a matrix manipulation method to eliminate the requirement of brick or
rectangle domain during constructing the eigenvalue and eigenvector of unconditional
covariance matrix.
This paper is arranged as follows. We first revisit the SLE that forms the
geostatistical inversion approach (section 2.2). Thereafter, the algorithm is
reformulated by collaborating with the data error, reduced order approach (i.e.
singular value decomposition, SVD), and irregular domain (section 2.3). Furthermore,
a perturbation method is proposed to improve the efficiency of covariance evaluation


process (section 2.4). In section 3, we test the proposed dual state-parameter
estimation algorithm with two synthetic examples and a tomographic survey at the
field site to demonstrate the superiority of the proposed method. Lastly, summary and
conclusions are presented.

**2.1. Groundwater Flow Model**.
The 2-D groundwater flow in heterogeneous confined aquifer can be described
as
$$\nabla \cdot [T(\mathbf{x}) \cdot \nabla h(\mathbf{x},t)] = S(\mathbf{x})\frac{\partial h(\mathbf{x},t)}{\partial t} \quad (1)$$

where $h$ is the head responses (m), $T$ is hydraulic transmissivity (m$^2$/day), $S$ is storage
coefficient (-), $\mathbf{x}$ is the vector in x and y directions, and $t$ represents time (day).

**2.2. Reduced order successive linear estimator**
The singular value decomposition (SVD) is employed to reduce the order of the
parameter covariance, leading to less memory requirement and more computational
efficiency inverse exercise. Afterward, the data error is considered to improve the
stability of convergence.
**(1) Hard Data**
When the hard data are available, kringing is used to estimate the conditional
parameter field and the corresponding conditional covariance matrix from the
measured parameters. It is expressed as
$$\hat{\mathbf{f}}^{(1)} = \hat{\mathbf{f}}^{(0)} + \boldsymbol{\varepsilon}_{ff}^{(0)}\mathbf{C}\mathbf{R}_{f^*f^*}^{-1}[\mathbf{f}^* - \hat{\mathbf{f}}^{(0)}] \quad (3)$$

and
$$\boldsymbol{\varepsilon}_{ff}^{(1)} = \boldsymbol{\varepsilon}_{ff}^{(0)} - \boldsymbol{\varepsilon}_{ff}^{(0)}\mathbf{C}\mathbf{R}_{f^*f^*}^{-1}\mathbf{C}^T\boldsymbol{\varepsilon}_{ff}^{(0)} \quad (4)$$



in which $\mathbf{f}^{*}$ ($n_m \times 1$) is the measured parameters, $\hat{\mathbf{f}}^{(0)}$ ($n_f \times 1$) is the unconditional
parameter field, $\hat{\mathbf{f}}^{(1)}$ ($n_f \times 1$) is the conditional parameter field. $n_f$ represents the
number of unknown parameters and $n_m$ represents the number of measured parameters.
$\boldsymbol{\varepsilon}_{ff}^{(0)}\mathbf{C}$ and $\mathbf{C}^{T}\boldsymbol{\varepsilon}_{ff}^{(0)}$ ($n_m \times n_f$) are the unconditional parameter covariance matrices
depicting the spatial correlation between the measured parameters and all parameters.
$\boldsymbol{\varepsilon}_{ff}^{(0)}$ ($n_f \times n_f$) is the unconditional parameter covariance matrix depicting the spatial
correlation between all parameters ($\hat{\mathbf{f}}$). $\mathbf{C}$ ($n_f \times n_m$) is a matrix eliminating the column
in $\boldsymbol{\varepsilon}_{ff}^{(0)}$ when the corresponding measured parameter is absent. $\boldsymbol{\varepsilon}_{ff}^{(1)}$ ($n_f \times n_f$) is the
conditional covariance marix of all parameters. The diagonal term of the matrix (i.e.,
residual variance) represents the remaining uncertainty of the estimated parameter
after the information (measurements) is included. A small residual variance indicates
the spatial trend of estimated parameter is close to the true, while a large value
indicates the estimate is close to the initial guessed value (i.e. heterogeneity is not
resolved). $\mathbf{R}_{f^{*}f^{*}}$ ($n_m \times n_m$) is the covariance matrix depicting the correlation between
measured parameters. Notice that Cholesky and QR decompositions are utilized to
solve the matrix multiplication of inverse $\mathbf{R}_{f^{*}f^{*}}$ when it is and is not a positive
definite matrix.

Since $n_f$ is usually huge, storage demand of $\boldsymbol{\varepsilon}_{ff}^{(0)}$ and $\boldsymbol{\varepsilon}_{ff}^{(1)}$ may not be always

affordable. Thus, singular value decomposition (SVD) is utilized to relieve this
memory burden by keeping the leading eigenvalues and eigenvectors. The SVD of
$\boldsymbol{\varepsilon}_{ff}$ is expressed as

$$\boldsymbol{\varepsilon}_{ff} = \mathbf{g}\,\boldsymbol{\lambda}\,\mathbf{g}^{T} \quad (5)$$

where $\boldsymbol{\lambda}$ ($n_{svd} \times n_{svd}$) is eigenvalues, $\mathbf{g}$ ($n_f \times n_{svd}$) is eigenvectors, and $n_{svd}$ is number



of leading eigenvalues. Substitute eq. (5) into eqs. (3) and (4), we have
$$\hat{\mathbf{f}}^{(1)} = \hat{\mathbf{f}}^{(0)} + \mathbf{g}^{(0)}\boldsymbol{\lambda}^{(0)}\mathbf{g}^{(0)T}\mathbf{C}\mathbf{R}_{ff}^{-1}[\mathbf{f}^* - \hat{\mathbf{f}}^{(0)}] \quad (6)$$
and
$$\mathbf{g}^{(1)}\boldsymbol{\lambda}^{(1)}\mathbf{g}^{(1)T} = \mathbf{g}^{(0)}\boldsymbol{\lambda}^{(0)}\mathbf{g}^{(0)T} - \mathbf{g}^{(0)}\boldsymbol{\lambda}^{(0)}\mathbf{g}^{(0)T}\mathbf{C}\mathbf{R}_{ff}^{-1}\mathbf{C}^T\mathbf{g}^{(0)}\boldsymbol{\lambda}^{(0)}\mathbf{g}^{(0)T} \quad (7)$$
Since $\mathbf{g}^{(1)}$ is always a function of $\mathbf{g}^{(0)}$, it can be expressed as
$$\mathbf{g}^{(1)} = \mathbf{g}^{(0)}\mathbf{u}^{(0)} \quad (8)$$
where is $\mathbf{u}^{(0)}$ ($n_{svd} \times n_{svd}$) is the matrix transferring the information of spatial
correlation of parameters to the next iteration. Accordingly, eq. (7) can reduce to
$$\mathbf{u}^{(0)}\boldsymbol{\lambda}^{(1)}\mathbf{u}^{(0)T} = \sqrt{\boldsymbol{\lambda}^{(0)}}(\mathbf{I} - \sqrt{\boldsymbol{\lambda}^{(0)}}\mathbf{g}^{(0)T}\mathbf{C}\mathbf{R}_{f^*f^*}^{-1}\mathbf{C}^T\mathbf{g}^{(0)}\sqrt{\boldsymbol{\lambda}^{(0)}})\sqrt{\boldsymbol{\lambda}^{(0)}} \quad (9)$$
in which $\mathbf{I}$ is an identity matrix. By decomposing eq. (9) with SVD, we obtain the
updated eigenvalue $\boldsymbol{\lambda}^{(1)}$ and $\mathbf{u}^{(0)}$. The updated eigenvector $\mathbf{g}^{(1)}$ can be evaluated
by eq. (8).
**(2) Error of Soft Data**
After considering the hard data, the inherently presented data errors (e.g.,
measurement, numeric, round-off, truncation errors, etc.) are included prior to the
parameter estimation. The estimated data error and the corresponding covariance
matrix are expressed as
$$\hat{\mathbf{e}}^{(r+1)} = \hat{\mathbf{e}}^{(r)} + \boldsymbol{\varepsilon}_{hh}^{(r)}[\mathbf{J}_{fh}^{(r)T}\boldsymbol{\varepsilon}_{ff}^{(r)}\mathbf{J}_{fh}^{(r)} + \boldsymbol{\varepsilon}_{hh}^{(r)}]^{-1}[\mathbf{h}^{(r)} - (\mathbf{h}^* + \hat{\mathbf{e}}^{(r)})] \quad (10)$$
and
$$\boldsymbol{\varepsilon}_{hh}^{(r+1)} = \boldsymbol{\varepsilon}_{hh}^{(r)} - \boldsymbol{\varepsilon}_{hh}^{(r)}[\mathbf{J}_{fh}^{(r)T}\boldsymbol{\varepsilon}_{ff}^{(r)}\mathbf{J}_{fh}^{(r)} + \boldsymbol{\varepsilon}_{hh}^{(r)}]^{-1}\boldsymbol{\varepsilon}_{hh}^{(r)} \quad (11)$$
in which $\mathbf{h}^*$ ($n_d \times 1$) is the observed head and $\mathbf{h}^{(r)}$ ($n_d \times 1$) is the simulated head
based on the estimated parameters from the $r^{\text{th}}$ iteration. $n_d$ represents the number of
measured state variables. The superscript $r$ is the iteration index starting from one.



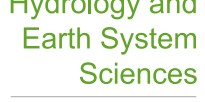

$\boldsymbol{\varepsilon}_{hh}^{(1)}$ ($n_d \times n_d$) is the unconditional covariance matrix of the observed head. The
diagonal terms represent the uncertainty of the measurement and the off diagonal
terms represent the correlation between errors. $\boldsymbol{\varepsilon}_{hh}^{(r)}$ and $\boldsymbol{\varepsilon}_{hh}^{(r+1)}$ ($n_d \times n_d$) are the
conditional covariance matrices. $\hat{\mathbf{e}}^{(1)}$ ($n_d \times 1$) is the initial data error. $\hat{\mathbf{e}}^{(r)}$ and $\hat{\mathbf{e}}^{(r+1)}$
($n_d \times 1$) are the estimated data error. $\mathbf{J}_{fh}^{(r)}$ ($n_f \times n_d$) is the sensitivity of observed head
with respect to the estimated parameters during the $r^{\text{th}}$ iteration.
The weight (i.e., $\mathbf{W}^{(r)} = \boldsymbol{\varepsilon}_{hh}^{(r)}[\mathbf{J}_{fh}^{(r)T} \boldsymbol{\varepsilon}_{ff}^{(r)} \mathbf{J}_{fh}^{(r)} + \boldsymbol{\varepsilon}_{hh}^{(r)}]^{-1}$) is a combination of observed
head covariance matrix ( $\boldsymbol{\varepsilon}_{hh}^{(r)}$ ) and simulated head covariance matrix
( $\mathbf{R}_{hh}^{(r)} = \mathbf{J}_{fh}^{(r)T} \boldsymbol{\varepsilon}_{ff}^{(r)} \mathbf{J}_{fh}^{(r)}$ ). It represents the ratio of data error ($\boldsymbol{\varepsilon}_{hh}^{(r)}$, including numeric and
measurement errors) to the total error ( $\mathbf{R}_{hh}^{(r)} + \boldsymbol{\varepsilon}_{hh}^{(r)}$, including model structure,
parameter, numeric, and measurement errors). When the model is poorly calibrated,
the simulated head based on the current model structure and parameter values is much
uncertain than that of observed head (i.e. $\mathbf{R}_{hh}^{(r)} \gg \boldsymbol{\varepsilon}_{hh}^{(r)}$). Thus, the weight ( $\mathbf{W}^{(r)}$ ) is
small and the algorithm trusts the observation ( $\mathbf{h}^{*}$ ) more than the prediction ( $\mathbf{h}^{(r)}$ ).
After assimilating the subsurface characteristic casted in the observation, the
uncertainty of simulated head ( $\mathbf{R}_{hh}^{(r)}$ ) reduces and the algorithm trusts the observation
( $\mathbf{h}^{*}$ ) less than the prediction ( $\mathbf{h}^{(r)}$ ). Therefore, dismiss between $\mathbf{h}^{*}$ and $\mathbf{h}^{(r)}$ are
reflected into $\hat{\mathbf{e}}^{(r)}$. This data error calibration step is similar to the Kalman filter, but
instead of using the observation from previous time step only, we consider all of the
available observation simultaneously.
Again, substitute eigenvalue $\boldsymbol{\lambda}$ and eigenvector $\mathbf{g}$ of $\boldsymbol{\varepsilon}_{ff}^{(r)}$ expressed in eq.
(5), the reduced order formulations of eqs. (10) and (11) are



$$\hat{\mathbf{e}}^{(r+1)} = \hat{\mathbf{e}}^{(r)} + \boldsymbol{\varepsilon}_{hh}^{(r)}[\mathbf{H}_{fh}^{(r)}\mathbf{H}_{fh}^{(r)T} + \boldsymbol{\varepsilon}_{hh}^{(r)}]^{-1}[\mathbf{h}^{(r)} - (\mathbf{h}^{*} + \hat{\mathbf{e}}^{(r)})] \quad (12)$$

and
$$\boldsymbol{\varepsilon}_{hh}^{(r+1)} = \boldsymbol{\varepsilon}_{hh}^{(r)} - \boldsymbol{\varepsilon}_{hh}^{(r)}[\mathbf{H}_{fh}^{(r)}\mathbf{H}_{fh}^{(r)T} + \boldsymbol{\varepsilon}_{hh}^{(r)}]^{-1}\boldsymbol{\varepsilon}_{hh}^{(r)} \quad (13)$$

where $\mathbf{H}_{fh}^{(r)}$ ($n_d \times n_{svd}$) is
$$\mathbf{H}_{fh}^{(r)} = \mathbf{J}_{fh}^{(r)T}\mathbf{g}^{(r)}\sqrt{\boldsymbol{\lambda}^{(r)}} \quad (14)$$

Notice that if the number of state ($n_d$) is huge, SVD can potentially be used to
decompose $\boldsymbol{\varepsilon}_{hh}^{(r)}$ (eq. 5) and reduce the storage requirement.
**(3) Soft Data.**
After estimating the data error, the measured state variables and data errors are
substituted into successive linear estimator (SLE) (Yeh et al., 1996) to estimate the
conditional parameter fields and the corresponding residual covariance matrix:
$$\hat{\mathbf{f}}^{(r+1)} = \hat{\mathbf{f}}^{(r)} + \boldsymbol{\varepsilon}_{ff}^{(r)}\mathbf{J}_{fh}^{(r)}[\mathbf{J}_{fh}^{(r)T}\boldsymbol{\varepsilon}_{ff}^{(r)}\mathbf{J}_{fh}^{(r)} + \boldsymbol{\varepsilon}_{hh}^{(r+1)}]^{-1}[(\mathbf{h}^{*} + \hat{\mathbf{e}}^{(r+1)}) - \mathbf{h}^{(r)}] \quad (15)$$

and
$$\boldsymbol{\varepsilon}_{ff}^{(r+1)} = \boldsymbol{\varepsilon}_{ff}^{(r)} - \boldsymbol{\varepsilon}_{ff}^{(r)}\mathbf{J}_{fh}^{(r)}[\mathbf{J}_{fh}^{(r)T}\boldsymbol{\varepsilon}_{ff}^{(r)}\mathbf{J}_{fh}^{(r)} + \boldsymbol{\varepsilon}_{hh}^{(r+1)}]^{-1}\mathbf{J}_{fh}^{(r)T}\boldsymbol{\varepsilon}_{ff}^{(r)} \quad (16)$$

The reduced order version of SLE can be derived by substitute eq. (5) into eqs.
(15) and (16). That is,
$$\hat{\mathbf{f}}^{(r+1)} = \hat{\mathbf{f}}^{(r)} + \mathbf{g}^{(r)}\sqrt{\boldsymbol{\lambda}^{(r)}}\mathbf{H}_{fh}^{(r)T}[\mathbf{H}_{fh}^{(r)}\mathbf{H}_{fh}^{(r)T} + \boldsymbol{\varepsilon}_{hh}^{(r+1)}]^{-1}[(\mathbf{h}^{*} + \hat{\mathbf{e}}^{(r+1)}) - \mathbf{h}^{(r)}] \quad (17)$$

and
$$\mathbf{g}^{(r+1)}\boldsymbol{\lambda}^{(r+1)}\mathbf{g}^{(r+1)T} = \mathbf{g}^{(r)}\boldsymbol{\lambda}^{(r)}\mathbf{g}^{(r)T} - \mathbf{g}^{(r)}\sqrt{\boldsymbol{\lambda}^{(r)}}\mathbf{H}_{fh}^{(r)T}[\mathbf{H}_{fh}^{(r)}\mathbf{H}_{fh}^{(r)T} + \boldsymbol{\varepsilon}_{hh}^{(r+1)}]^{-1}\mathbf{H}_{fh}^{(r)}\sqrt{\boldsymbol{\lambda}^{(r)}}\mathbf{g}^{(r)T} \quad (18)$$

Using eq. (8), eq. (18) further reduces to
$$\mathbf{u}^{(r)}\boldsymbol{\lambda}^{(r+1)}\mathbf{u}^{(r)T} = \sqrt{\boldsymbol{\lambda}^{(r)}}(\mathbf{I} - \mathbf{H}_{fh}^{(r)T}[\mathbf{H}_{fh}^{(r)}\mathbf{H}_{fh}^{(r)T} + \boldsymbol{\varepsilon}_{hh}^{(r+1)}]^{-1}\mathbf{H}_{fh}^{(r)})\sqrt{\boldsymbol{\lambda}^{(r)}} \quad (19)$$

By decomposing eq. (19) with SVD, we can evaluate the updated eigenvalue $\boldsymbol{\lambda}^{(r+1)}$





and $\mathbf{u}^{(r)}$. The updated eigenvector $\mathbf{g}^{(r+1)}$ then can be calculated by eq. (8).
**(4) Convergence Criterion**
The estimated field is considered as the converge one when the spatial variance
of the estimated parameter duing several iterations are steady. The tolerance using
mean squared error between the observed and simulated states is no longer necessary.

**2.3. Required Inputs**
To initiate the algorithm, the initial guess of parameter field $\hat{\mathbf{f}}^{(0)}$ and data error
$\hat{\mathbf{e}}^{(0)}$, as well as the unconditional covariance matrix of the parameters $\mathbf{\varepsilon}_{ff}^{(0)}$ and
observed data $\mathbf{\varepsilon}_{hh}^{(0)}$ are required. The details are explained as followings:
**Parameter Field:** The initial parameter field $\hat{\mathbf{f}}^{(0)}$ can be any reasonable values
based on the prior knowledge.
**Parameter Covariance:** We assume the unconditional parameter covariance
matrix is defined by an exponential covariance function
$$\mathbf{\varepsilon}_{ff}^{(0)} = Var \cdot \exp\left( \frac{-\left|\mathbf{d_x}\right|}{\lambda_x} + \frac{-\left|\mathbf{d_y}\right|}{\lambda_y} \right) \quad (20)$$

where $Var$ represents the unconditional spatial variance of the parameter; $\mathbf{d_x}$ ($n_f \times 1$)
and $\mathbf{d_y}$ ($n_f \times 1$) are the distance between two parameters in $x$ and $y$ directions; $\lambda_x$
and $\lambda_y$ are the correlation lengths (m) in $x$ and $y$ directions.
The reduced order algorithm requires the evaluation of unconditional parameter
covariance matrix $\mathbf{\varepsilon}_{ff}^{(0)}$ in terms of eigenvalue $\mathbf{\lambda}^{(0)}$ and eigenvector $\mathbf{g}^{(0)}$. In the
real-world problem, the number of parameters $n_f$ is usually in the order of $10^3$ to $10^5$,
and the computational cost of conducting full SVD is $n_f^3$ ($O(n_f^3)$). Alternately,



truncated SVD with the complexity in $O(n_{ft}^2 n_f)$ can be used to approximate the
original eigenvalue and eigenvector. $n_{ft}$ is the number of randomly chose column in
$\varepsilon_{ff}^{(0)}$.

In addition to the numeric approach, the analytical solution of eigenvalues $\lambda_n$

and eigenvectors $\mathbf{g}_n$ with brick grid and domain (Ghanem and Spanos, 2003; Zhang
and Lu, 2004) is also available. In 2-D domain, they are analytically express as
$$\lambda_n = Var \frac{2\lambda_x}{\lambda_x^2 w_{n,x}^2 + 1} \frac{2\lambda_y}{\lambda_y^2 w_{n,y}^2 + 1} \quad (21)$$

$$\mathbf{g}_n = \frac{\lambda_x w_{n,x} \cos(w_{n,x}x) + \sin(w_{n,x}x)}{\sqrt{\frac{(\lambda_x^2 w_{n,x}^2 + 1)L_x}{2} + \lambda_x}} \frac{\lambda_y w_{n,y} \cos(w_{n,y}y) + \sin(w_{n,y}y)}{\sqrt{\frac{(\lambda_y^2 w_{n,y}^2 + 1)L_y}{2} + \lambda_y}} \quad (22)$$

where $w_{n,x}$ and $w_{n,y}$ are the positive roots of the characteristic equations
$$(\lambda_x^2 w_{n,x}^2 - 1)\sin(w_{n,x}L_x) = 2\lambda_x w_{n,x} \cos(w_{n,x}L_x) \quad (23)$$

and
$$(\lambda_y^2 w_{n,y}^2 - 1)\sin(w_{n,y}L_y) = 2\lambda_y w_{n,y} \cos(w_{n,y}L_y) \quad (24)$$

where $L_x$ and $L_y$ are the width of model domain in $x$ and $y$ directions.

Notice that if the model domain is irregular (i.e., not a line, squared, or brick

shape), one can first construct the eigenvalue and eigenvector for a regular domain
whose size is greater than the irregular one. Afterward, the eigenvector of the irregular
domain can be evaluated by
$$\mathbf{g}_{irreg} = \mathbf{C}_2 \mathbf{g}_{reg} \quad (25)$$

in which $\mathbf{C}_2$ ($n_{f,irreg} \times n_{f,reg}$) is a matrix to eliminate the rows of $\mathbf{g}_{reg}$ if the
corresponding grids are outside the model domain; $n_{f,reg}$ is the number of parameter of



the regular line, squared, or brick domain; $n_{f,irreg}$ is the number of parameter of the
irregular domain. $\mathbf{g}_{reg}$ and $\mathbf{g}_{irreg}$ are the eigenvectors of regular and irregular
domains.
**Data Error:** The initial data error $\hat{\mathbf{e}}^{(0)}$ can set as zero.
**Data Covariance:** The unconditional covariance matrix of the observed data
$\boldsymbol{\varepsilon}_{hh}^{(0)}$ is a diagonal matrix if the data error are mutually independent. Otherwise, a
covariance function (e.g., eq. (20)) can be utilized to describe the unconditional
correlation.

**2.4. Evaluation of Covariance**
The algorithm also requires the evaluation of squared root of cross-covariance
$\mathbf{H}_{fh}^{(r)}$. One can evaluate the sensitivity by adjoint approach (e.g., Sykes et al., 1985;
Sun and Yeh, 1990) first and substitute it into eq. (14) to derive $\mathbf{H}_{fh}^{(r)}$. The
computational cost of the adjoint approach is to run the linear adjoint forward model
$n_w$ (number of observation wells) to $n_d$ (number of states) times, depending on the
model configurations (e.g., confined, unconfined, saturated, unsaturated, and the types
of boundary condition, etc.).
On the other hand, a perturbation approach (e.g., forward, backward, central
differences, etc.) can be utilized to directly evaluate $\mathbf{H}_{fh}^{(r)}$ so that the computation of
sensitivity is eliminated. Let $G(\cdot)$ represent the groundwater flow governing
equation and its Taylor expansion evaluated on $\hat{\mathbf{f}}^{(r)} + \mathbf{g}^{(r)}\delta$ is
$$G(\hat{\mathbf{f}}^{(r)} + \mathbf{g}^{(r)}\delta) = G(\hat{\mathbf{f}}^{(r)}) + G'(\hat{\mathbf{f}}^{(r)})\mathbf{g}^{(r)}\delta + G''(\hat{\mathbf{f}}^{(r)})\frac{(\mathbf{g}^{(r)}\delta)^2}{2} + G'''(\hat{\mathbf{f}}^{(r)})\frac{(\mathbf{g}^{(r)}\delta)^3}{3!} + ... \quad (26)$$
$\delta$ is an arbitrary value controlling the accuracy of approximation. Manipulating eq.


(26) yields
$$G'(\hat{\mathbf{f}}^{(r)})\mathbf{g}^{(r)} = \mathbf{J}_{fh}^{(r)T}\mathbf{g}^{(r)} = \frac{G(\hat{\mathbf{f}}^{(r)}+\mathbf{g}^{(r)}\delta)-G(\hat{\mathbf{f}}^{(r)})}{\delta} - G''(\hat{\mathbf{f}}^{(r)})\frac{(\mathbf{g}^{(r)})^2\delta}{2} - G'''(\hat{\mathbf{f}}^{(r)})\frac{(\mathbf{g}^{(r)})^3\delta^2}{3!} - ...$$

(27)

Multiplying both sides with $\sqrt{\lambda^{(r)}}$, eq. (27) becomes
$$\mathbf{H}_{fh}^{(r)} = \left[\frac{G(\hat{\mathbf{f}}^{(r)}+\mathbf{g}^{(r)}\delta)-G(\hat{\mathbf{f}}^{(r)})}{\delta} - G''(\hat{\mathbf{f}}^{(r)})\frac{(\mathbf{g}^{(r)})^2\delta}{2} - G'''(\hat{\mathbf{f}}^{(r)})\frac{(\mathbf{g}^{(r)})^3\delta^2}{3!} - ...\right]\sqrt{\lambda^{(r)}}    (28)$$
Accordingly, $\mathbf{H}_{fh}^{(r)}$ can be approximated by
$$\mathbf{H}_{fh}^{(r)} \approx \frac{G(\hat{\mathbf{f}}^{(r)}+\mathbf{g}^{(r)}\delta)-G(\hat{\mathbf{f}}^{(r)})}{\delta}\sqrt{\lambda^{(r)}}    (29)$$
and the corresponding error is
$$err = \sqrt{\lambda^{(r)}}G''(\hat{\mathbf{f}}^{(r)})\frac{(\mathbf{g}^{(r)})^2\delta}{2} + ...    (30)$$
To evaluate $\mathbf{H}_{fh}^{(r)}$, we need to run the forward model $n_{svd}$ (number of kept eigens) $\times$
$n_{event}$ (number of pumping or injection events) times.
If we further evaluate $G(\cdot)$ on $\hat{\mathbf{f}}^{(r)} - \mathbf{g}^{(r)}\delta$ and combine it with $G(\hat{\mathbf{f}}^{(r)}+\mathbf{g}^{(r)}\delta)$,
we have
$$G(\hat{\mathbf{f}}^{(r)}+\mathbf{g}^{(r)}\delta) - G(\hat{\mathbf{f}}^{(r)}-\mathbf{g}^{(r)}\delta) = 2\left[G'(\hat{\mathbf{f}}^{(r)})\mathbf{g}^{(r)}\delta + G'''(\hat{\mathbf{f}}^{(r)})\frac{(\mathbf{g}^{(r)}\delta)^3}{3!} + ...\right]    (31)$$
Multiplying both sides with $\sqrt{\lambda^{(r)}}$ and $\mathbf{H}_{fh}^{(r)}$ can be approximated by
$$\mathbf{H}_{fh}^{(r)} \approx \frac{G(\hat{\mathbf{f}}^{(r)}+\mathbf{g}^{(r)}\delta)-G(\hat{\mathbf{f}}^{(r)}-\mathbf{g}^{(r)}\delta)}{2\delta}\sqrt{\lambda^{(r)}}    (32)$$
The corresponding error is
$$err = \sqrt{\lambda^{(r)}}G'''(\hat{\mathbf{f}}^{(r)})\frac{(\mathbf{g}^{(r)})^3\delta^2}{3!} + ...    (33)$$
The evaluation of more accurate $\mathbf{H}_{fh}^{(r)}$ requires the exercise of forward model $2n_{svd} \times$
$n_{event}$ times.






**2.5. Computational Advantages**

The proposed reduced-order dual state-parameter inverse algorithm is efficient
when the number of kept leading eigens ($n_{svd}$) is less than 1500. If the ratio of domain
size and correlation length is huge, large $n_{svd}$ value increase the computational cost of
SVD ($O(n_{kl}^3)$). Furthermore, evaluating $\mathbf{H}_{fh}^{(r)}$ through the forward or backward finite
difference approach is efficient for many types of forward models (e.g., variable
saturated diffusion equation, advection diffusion equation). It only requires executing
the forward model for $n_{svd} \times n_{event}$ (number of pumping events) times. On the contrary,
when the forward model is elegant (e.g., fully saturated diffusion equation), it is
cost-effective to evaluate the sensitivity of state with respect to unknown parameter
(eq. 14, $\mathbf{J}_{fh}^{(r)}$) through the adjoint method. Only $n_w$ (number of observation wells)
forward runs is required. In addition, updating state variable errors is efficient when
$n_d$ (number of state variable) is less than 10000. The most expensive additional
computational cost is to solve the inverse $n_d \times n_d$ matrix (eqs. 12 and 13) through either
Cholesky or QR decompositions (matrix multiplication is an easy task under the
parallel computing scheme).

**3. Algorithm Verification**

In this section, three cases are used to examine the robustness of the proposed
algorithm. The first and second cases involve hydraulic tomographic surveys in a
synthetic aquifer without and with observation error, respectively. The third case is a
2-D application of tomography experiment in the field site.
The coefficient of determination ($R^2$) and the mean squared error (i.e., $L_2$ norm),
defined as



$$R^2 = \left[ \frac{(\mathbf{f}^* - \overline{\mathbf{f}}^*)^T (\hat{\mathbf{f}}^{(r)} - \overline{\hat{\mathbf{f}}}^{(r)})}{n_f \, std(\mathbf{f}^*) std(\hat{\mathbf{f}}^{(r)})} \right]^2 \quad (34)$$

and
$$L_2 = \frac{(\mathbf{f}^* - \hat{\mathbf{f}}^{(0)})^T (\mathbf{f}^* - \hat{\mathbf{f}}^{(0)})}{n_f} \quad (35)$$

are utilized to evaluate the similarity between the reference and estimated parameter
fields. Overbar represents the average. $std(\cdot)$ stands for the standard deviation.

**3.1. Observation-Error Free Synthetic Case**

The observation-error free synthetic case considers transient state HT in a
two-dimensional horizontal confined aquifer of 30×30 square elements (figure 1).
Each element is 1 (m)×1 (m). The aquifer is bounded by the constant head boundary
(30 m). The initial head is uniform (30 m) everywhere.

**(a) Forward Model**

The reference field (figure 1) is generated using a spectral method (Gutjahr, 1989;
Robin et al., 1993) with mean geometric $T$ of one (m²/day), variance of ln$T$ of one (-),
and correlation scales of 10 (m) at both x and y directions. Eight wells (white dots)
are evenly installed in the aquifer to collect the aquifer responses induced by three
sequential pumping tests from early time till the system reaches steady state. The
pumping wells are labeled with squares. The noise free observed heads only contain
the numerical error (e.g., round-off and truncation errors), and its value is smaller than
$10^{-7}$ (m). The $S$ is a constant value of 0.001 (-). The initial time step is 0.001 (day) and
the maximum time step is 1 (day).

**(b) Inverse Model**

Assume $S$ is known and we would like to estimate the spatial $T$ distribution. The





initial mean $T$ ($\hat{\mathbf{f}}^{(0)}$) is one (m$^2$/day), variance of $\ln T$ ($\boldsymbol{\varepsilon}_{ff}^{(0)}$) is one (-), variance of
observed head ($\boldsymbol{\varepsilon}_{hh}^{(0)}$) is $10^{-4}$ (m$^2$), and the correlation lengths $\lambda_x$ and $\lambda_y$ are 10 (m).
(c) **Results of Estimate**
Figure 2 shows the performances of the estimated $T$ value using old algorithm
(SLE) and figure 3 presents the performances using the new algorithm. Figure 2a
presents the evolutions of mean squared error between the observed and simulated
heads ($L_2$ norm) and the spatial variance of $\ln T$ (Var $\ln T$) during the calibration
process. Figure 2b is the calibrated head at the final iteration. Figures 2c and 2d are
the final and best estimated $T$ field, respectively. Figure 2e and f are the scatter plots
of the estimated $T$ verses reference $T$ corresponding to the final (figure 2c) and best
(figure 2d) estimated $T$ fields. As displayed in figure 2a, after $L_2$ norm approaches
steady, the spatial variance of estimated $T$ (pink line) still increases with a constant
rate. The gaining of spatial variation of estimated $T$ values comes from the over
calibrated observed head. Due to the natural of least squared algorithm (e.g.,
minimizing the mean squared error of state variables), the algorithm compensates the
numeric errors by adjusting the estimated $T$ to unreasonably high and low values,
although the general spatial trend of the estimated $T$ fields remains similar. As the
result, compare to the best estimate of $T$ field (figure 2d and f), the final estimate
diverges (figure 2c and e).
On the contrary, estimate using the new algorithm does not encounter the
divergence issue. As shown in the calibration process (figure 3a), the spatial variance
of estimated $T$ (pink line) reaches steady after $L_2$ decays to the value of $10^{-10}$ (i.e.,
magnitude of numeric error). The final estimated field (figure 3c) converges and the
performances in terms of the statistical indices (figure 3d), namely $L_2$, $R^2$
(determination coefficient), or the slope and intercept of the fitted linear relationship



between the estimates and the true values, are equally good as the best estimate by the
previous algorithm (figure 2c). In other word, the new algorithm eliminates the over
fitting issue.

**3.2. Noisy Synthetic Case**
This example aims to reveal the advantages of the algorithm when the
measurement errors are presented. To accomplish this goal, the Gaussian noises with
standard deviation of $10^{-3}$ (m) are superimposed on the observed heads discussed in
section 3.1. The design of inverse model is identical with those explained in section

3.1.

Figure 4 shows the performances (evolution of calibration process, head fitting,
contour of the estimate field, and the scatter plot between the estimate and reference
fields) of the estimated $T$ value using original algorithm, and figure 5 presents the
performances using the new one. By comparing the final estimate with the manually
selected best estimate of original SLE (figure 4d and f), the final estimated $T$ field
diverges as indicated by the increase in variance of $\ln T$ (pink line in figure 4a),
unreasonable high and low values (red and blue spots in figure 4c) of the final
estimated $T$ fields, and the uncorrelated estimate and reference $\ln T$ values (figure 4e).
On the contrary, the final estimate using the proposed algorithm shows that the
estimated field converges to the reasonable spatial pattern and values. The variance of
$\ln T$ (pink line in figure 5a) approaches stable and the simulated heads reproduce the
adjusted observed heads (sum of observed heads and estimated head errors, figure 5b).
Furthermore, the contour map and scatterplot of the final estimate (figure 5c and d)
suggest the estimated field is close to the manually selected best estimate of original
SLE (figure 4d and f) and the reference (figure 1). This means the new method no
longer overestimate the parameter fields and can automatically converge to an optimal



estimate under the given constrains.

**3.3. Field Data**

The proposed algorithm is applied to a river stage tomographic survey conducted

in Pingtun Plain, Taiwan. It is a 1200 km$^2$ catchment with three major rivers
penetrating from the north to south (figure 6). The plain is bound by foothills and
river valleys at the north, faults at the west and east, and the shoreline at the south. As
illustrated in figure 6b, the geology inferred from well logs shows that the upstream
subsurface is consist of gravel. Follows by the layered sand and clay structure at
middle and down streams. The regions with unconsolidated coarse sediments (gravel
and sand) are aquifer and with fine sediments (silt and clay) are aquitard. The aquitard
is characterized as marine deposition because abundant fossils such as shells and
foraminifera live in the shallow marine and lagoon are discovered. The aquifer is
characterized as non-marine deposition. Figure 6c presents the stream stage and
groundwater level variations during 2006. The average annual rainfall is 2500 mm,
with most of the precipitation happen between May and September.

We focus on characterizing the heterogeneity of shallow aquifer because it is the

major water source of agriculture, industrial, and municipal water supply. The average
aquifer thickness is 40 m. This catchment is discretized into a two-dimensional
horizontal confined aquifer with 5619 elements. Each element is 0.5 (km)×0.5 (km).
There are 36 monitoring wells evenly placed across the catchment and measuring the
hourly groundwater level variation of the aquifer since 1998. The aquifer is bounded
by the time varying head boundary. The time varying heads along the boundary are
extrapolated by kriging using the observed head collected from all of the monitoring
wells. Water levels collected from stream gauges are incorporated into the diffusion
wave equation to estimate the stream stages along the river. These estimated stages



are then treated as the prescribed head in the groundwater model. The initial
groundwater level is estimated by spinning up the model for 6 years prior to June
2006 utilizing the effective $T$ (1 (m$^2$/day)), effective $S$ (10$^{-5}$ (-)), time varying head
boundary, and stream stage variations.

The denoised groundwater levels from June to September 2006 are selected

using the strategy (i.e., wavelet) discussed in Wang et al. (2017). There is a total of
1440 measured heads selected for river stage tomograhic survey. The initial mean $T$
($\hat{\mathbf{f}}^{(0)}$) is 1 (m$^2$/day), variance of ln$T$ ($\varepsilon_{ff}^{(0)}$) is one (-), variance of observed head ($\varepsilon_{hh}^{(0)}$)
is 10$^{-4}$ (m$^2$), and the correlation lengths $\lambda_x$ and $\lambda_y$ are 15 (km). For simplicity, we
assume $S$ is uniform and focus on estimating the spatial $T$ distribution. The patterns of
estimated $T$ fields should be consistent with the hydraulic diffusivity field.

Figure 7 presents the calibration using the original SLE algorithm. The

increasing of variance of ln$T$ (figure 7a) near the end of iteration (iteration 100)
suggests the estimate diverges, although the parameter field reproduces the observed
drawdowns (figure 7b). The unreasonable huge spatial $T$ variation corresponds to the
significantly low and high values on the contour map of the final estimate field (figure
7c). The contour map of manually selected best estimate is shown in figure 7d and the
calibrated heads are similar with those in figure 7b.

Figure 8 shows the estimate using the new algorithm. The performance clearly

demonstrates the robustness and usefulness of the new algorithm on characterizing the
subsurface heterogeneity. Compared with the variance of ln$T$ in figure 7a, it stabilizes
at the end of iteration (figure 8a) while still reproduces the adjusted observed heads
(figure 8b). The estimated field (figure 8c) shares the similar spatial patterns with the
manually selected one (figure 7d).

To further examine the reliability of the estimate, the estimated $T$ field is



compared with the map of geological sensitivity regions (figure 9) delineated by the
Department of Central Geology Survey, Taiwan. The geological sensitivity regions
represent the major areas water recharges the aquifer. They are categorized by core
samples, geophysics (e.g., electric resistivity), and geochemical survey. In general, the
deposition of geological sensitivity region is gravel and the aquifer thickness is
greater than 100 m (blue areas in figure 6b). Compared figure 9 with figure 7d and 8c,
the high $T$ regions located near the upper streams (red areas) are in parallel with the
geological sensitivity regions.

**4. Conclusion**
In this paper, a reduced order geostatistical model is developed to account for the
subsurface heterogeneity. This method includes the evaluation of the errors of state
variables and unknown parameters to improve the robustness of convergence. The
over fitting problem (i.e., diverged estimated parameter fields) is leveraged by
considering these errors into the calibration process. The memory burden (i.e. high
dimensional parameter covariance) and requirement of domain shape (e.g., brick or
rectangle) are also relieved by approximating the parameter covariance matrix
through limited number of leading eigenvalues and eigenvectors using SVD.
Meanwhile, the computation of sensitivity is replaced by the direct evaluation of
cross-covariance through the finite differencing method. The modification relaxes
barrier of implementing this inverse algorithm to different disciplines because the
derivation of adjoint state method is no longer necessary. Lastly, as the stability of
convergence is robust and the evaluation of cross-covariance (sensitivity) is efficient,
the proposed algorithm is valuable and attractive for multi-discipline scientific
problems, especially useful and convenient for assimilating different types of
measurements.




**5. Acknowledgment**

This research is in part by U.S. Civilian Research and Development Foundation (CRDF Global) under the award number (DAA2-15-61224-1): Hydraulic tomography in shallow alluvial sediments: Nile River Valley, Egypt. The authors gratefully acknowledge the financial support by Minster of Science and Technology, Taiwan under award number 108-2116-M-002-029-MY3.

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



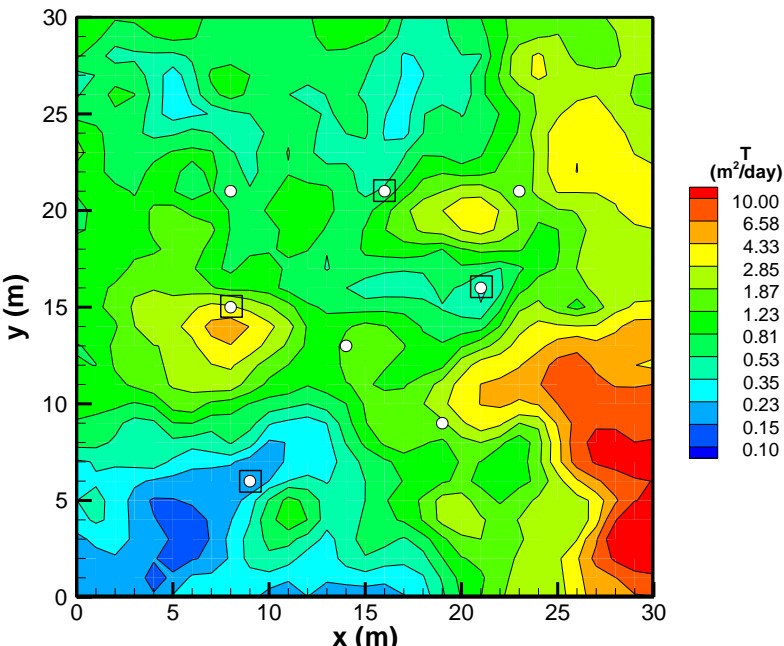


Figure 1. Reference hydraulic transimissivity $T$ (m$^2$/day) field. The white dots
represent monitoring wells and the squared are pumping wells. Four boundaries are
the constant head.



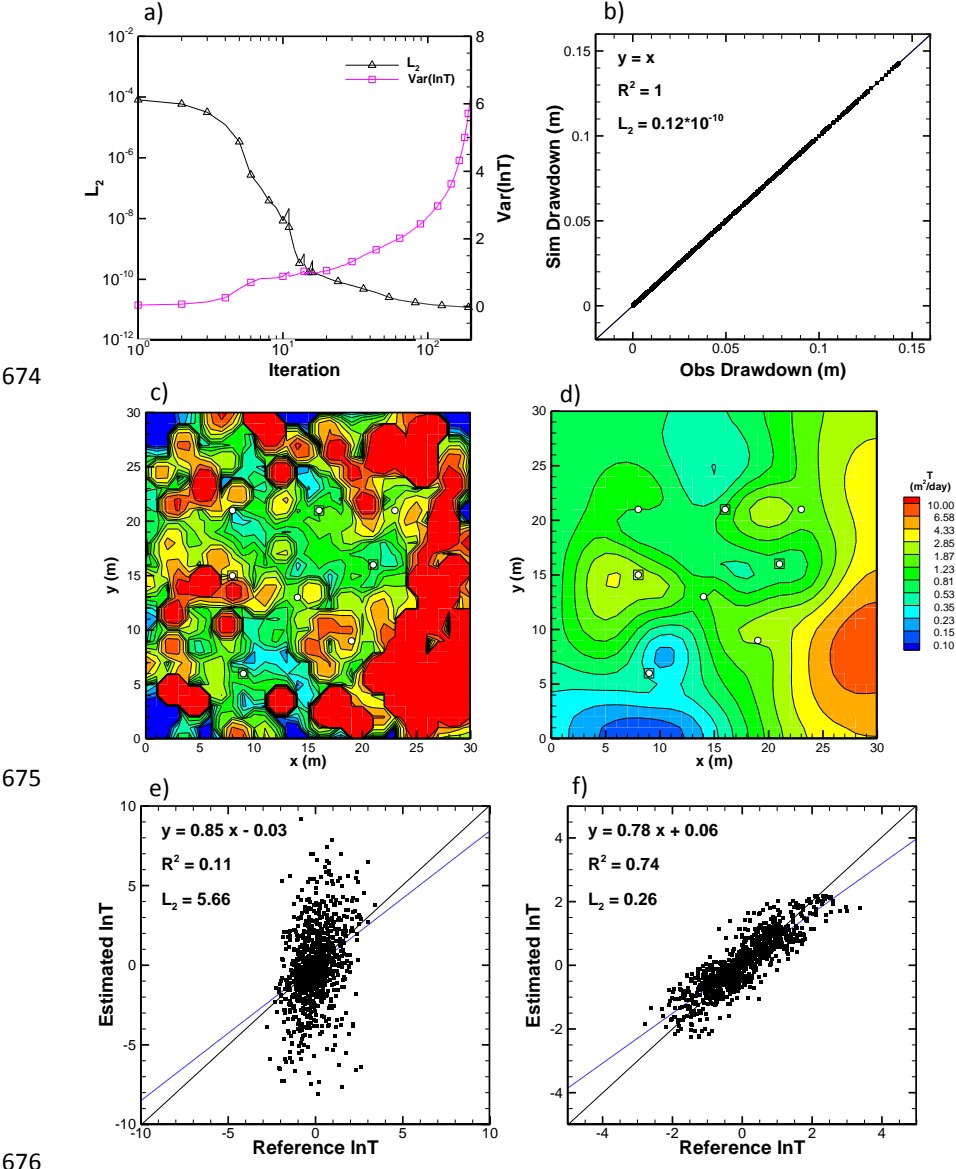




Figure 2. The estimated hydraulic transmissivity $T$ (m$^2$/day) field using noise free
observed head and old algorithm. a) The evolutions of mean squared error between
the observed and simulated heads ($L_2$ norm) and the spatial variance of ln$T$ (Var ln$T$)
during the calibration process. b) The calibrated head of the final iteration. c) The
final estimated $T$ field. d) The best estimated $T$ field. e) The scatter plots of final
estimated verses reference ln$T$. f) The scatter plots of best estimated verses reference
ln$T$.




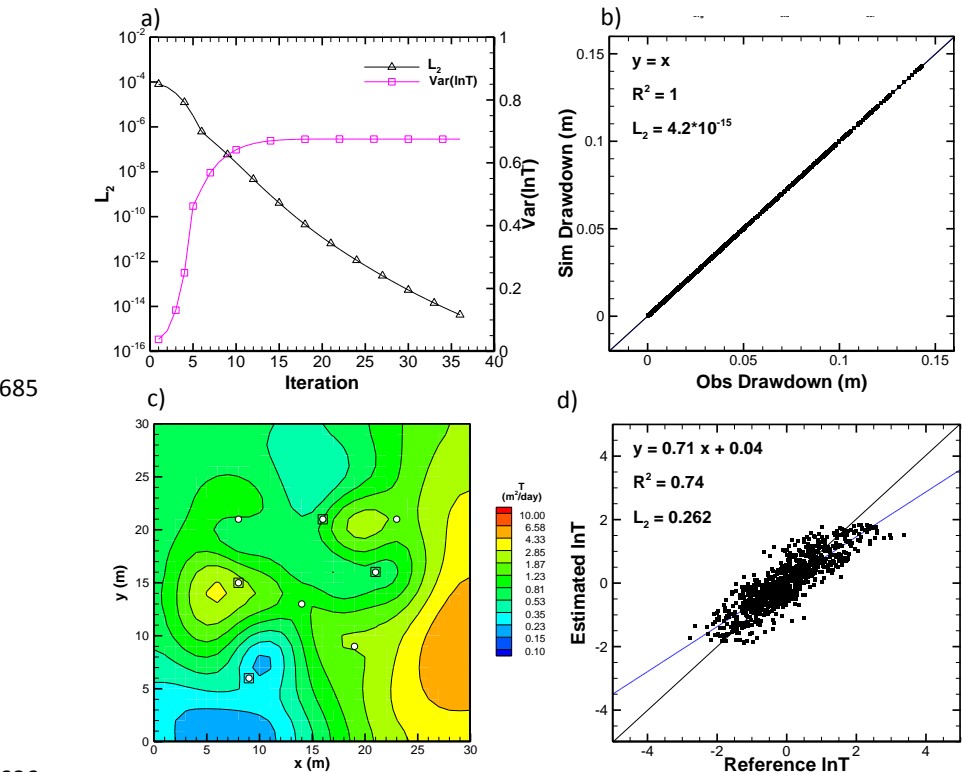

Figure 3. The estimated hydraulic transmissivity $T$ (m$^2$/day) field using noise free
observed head and new algorithm. a) The evolutions of mean squared error between
the observed and simulated heads ($L_2$ norm) and the spatial variance of ln$T$ (Var ln$T$)
during the calibration process. b) The calibrated head of the final iteration. c) The
final estimated $T$ field. d) The scatter plots of final estimated verses reference ln$T$.


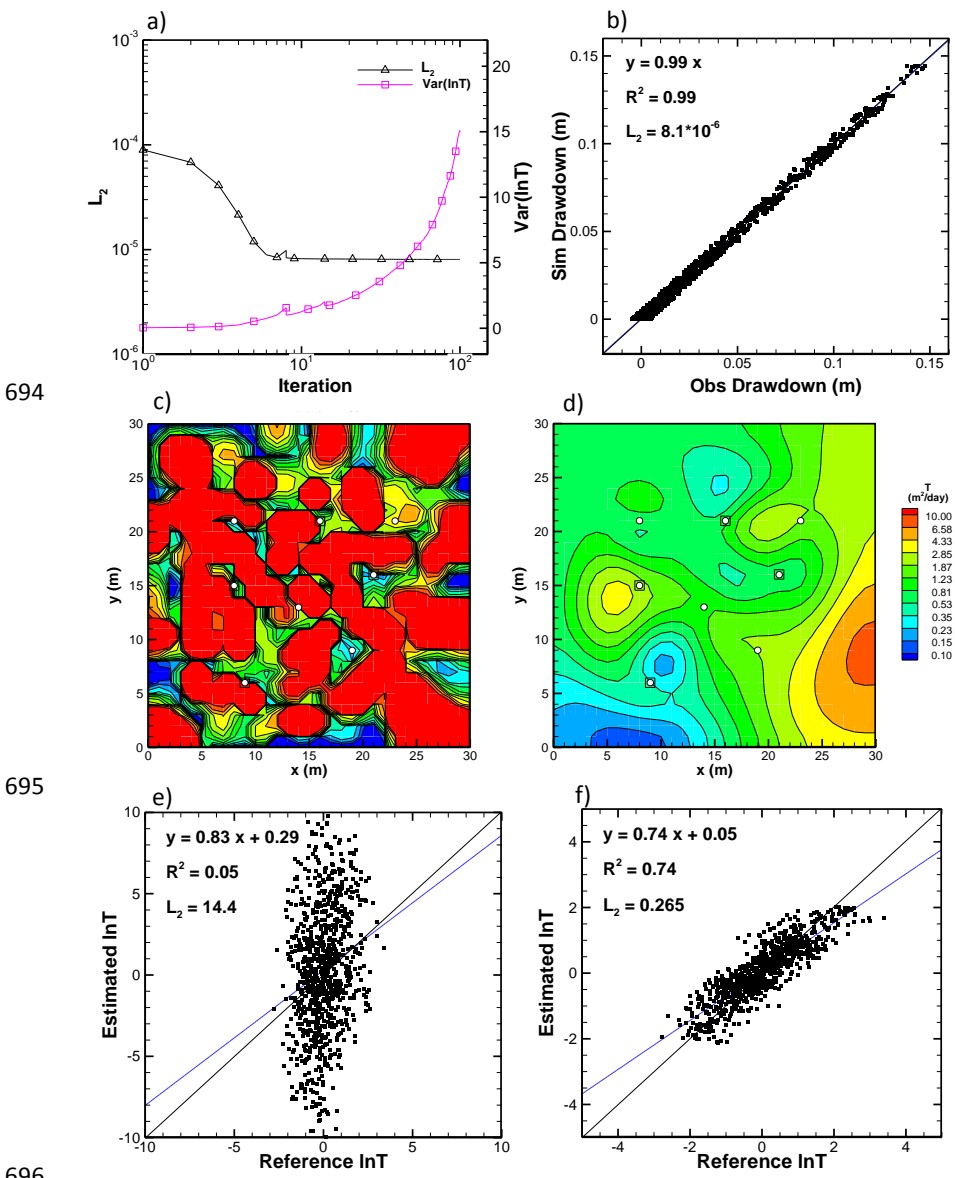

Figure 4. The estimated hydraulic transmissivity $T$ (m$^2$/day) field using noisy
observed head and old algorithm. a) The evolutions of mean squared error between
the observed and simulated heads ($L_2$ norm) and the spatial variance of ln$T$ (Var ln$T$)
during the calibration process. b) The calibrated head of the final iteration. c) The
final estimated $T$ field. d) The best estimated $T$ field. e) The scatter plots of final
estimated verses reference ln$T$. f) The scatter plots of best estimated verses reference
ln$T$.



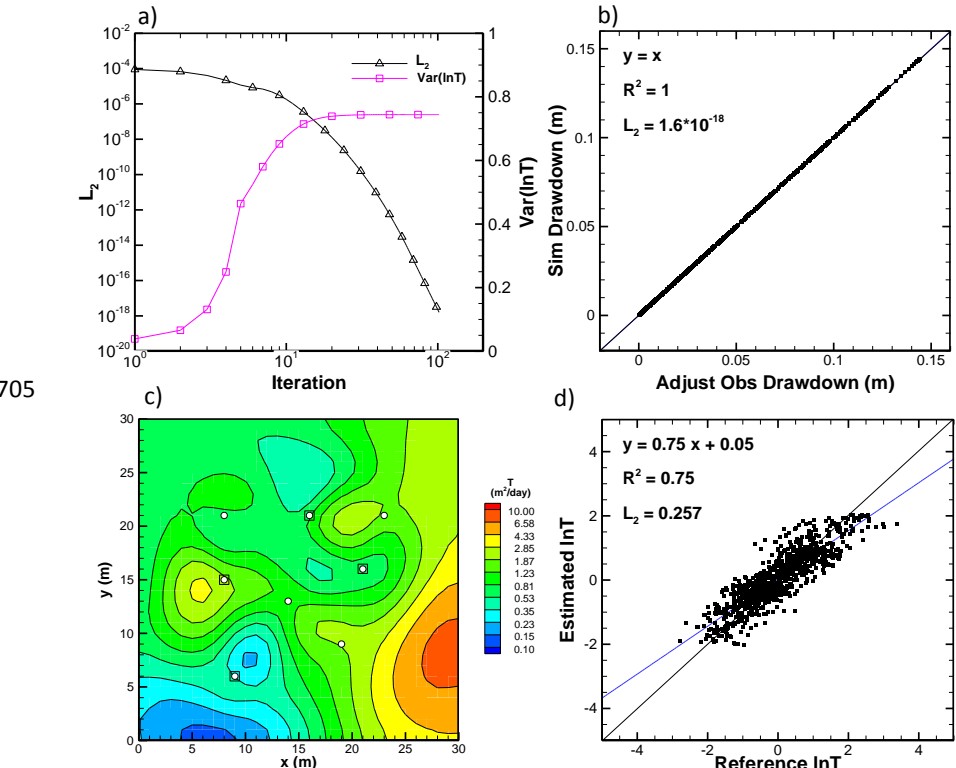

Figure 5. The estimated hydraulic transmissivity $T$ (m²/day) field using noisy
observed head and new algorithm. a) The evolutions of mean squared error between
the observed and simulated heads ($L_2$ norm) and the spatial variance of ln$T$ (Var ln$T$)
during the calibration process. b) The calibrated heads verse adjusted observed heads
(observed head + estimated error) of the final iteration. c) The final estimated $T$ field.
d) The scatter plots of final estimated verses reference ln$T$.



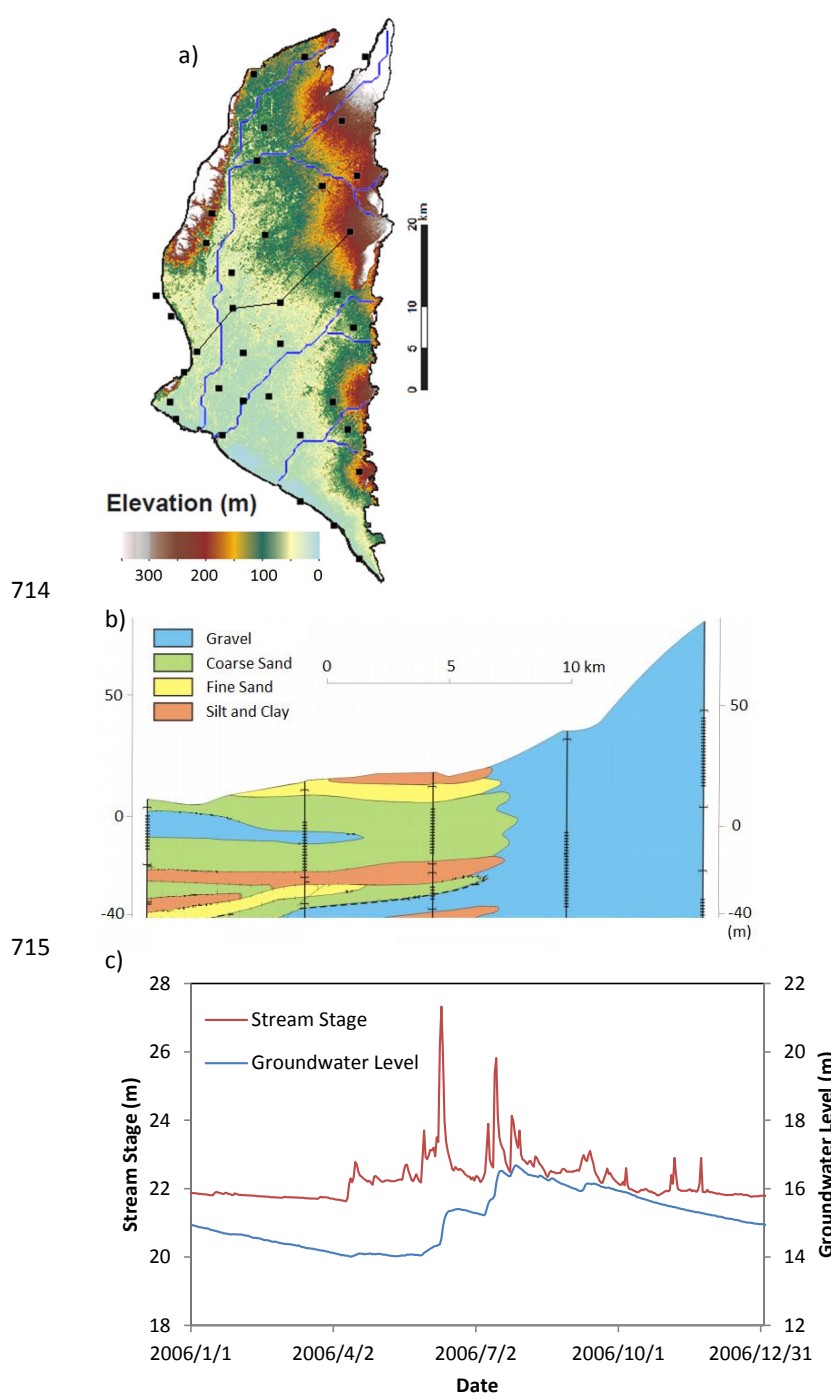







Figure 6. a) Topography of the study plain. The blue lines represent rivers and the
black rectangles are groundwater monitoring wells. The black line is geological cross
section. b) Geological cross section. c) Stream stage and groundwater level variations
during 2006. a) and b) are modified from the website of Water Resources Agency, the
administrative agency of the Ministry of Economic Affairs in Taiwan.



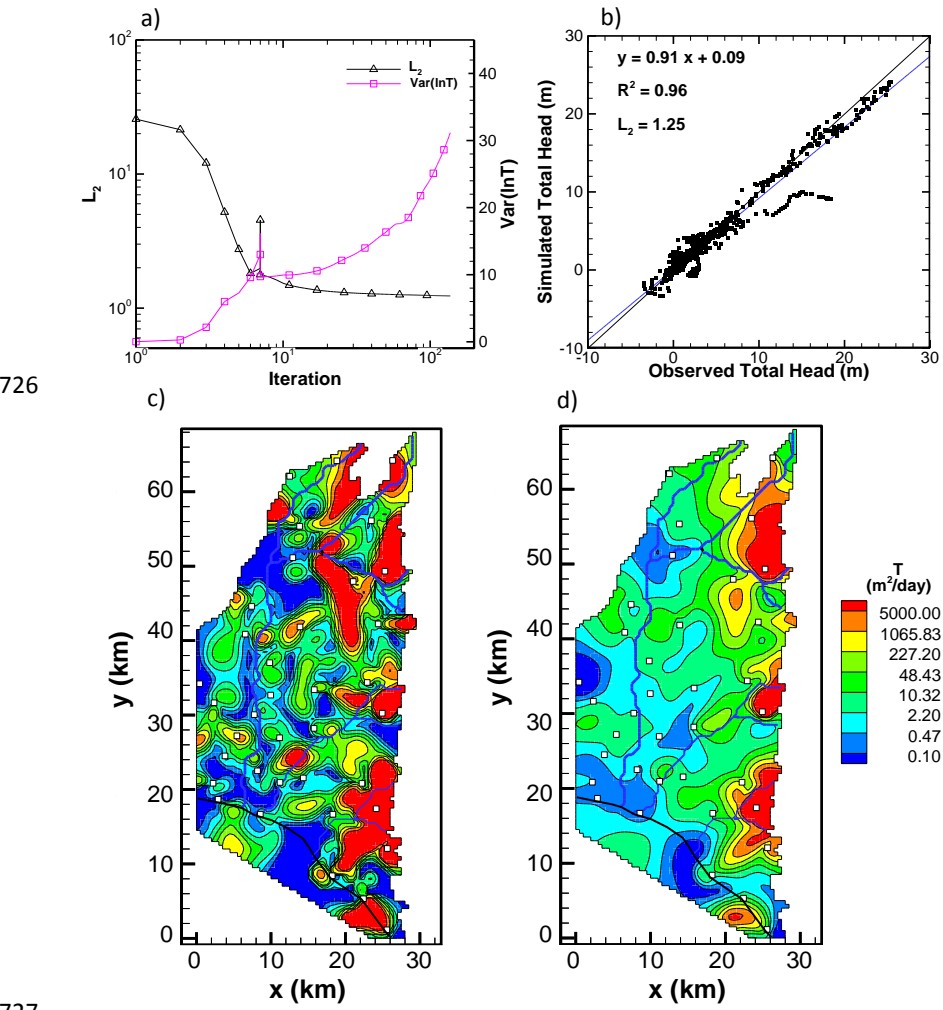



Figure 7. The estimated hydraulic transmissivity $T$ (m²/day) field using observed head
in the field and old algorithm. a) The evolutions of mean squared error between the
observed and simulated heads ($L_2$ norm) and the spatial variance of ln$T$ (Var ln$T$)
during the calibration process. b) The calibrated head of the final iteration. c) The
final estimated $T$ field. d) The best estimated $T$ field. The white squares represent
wells, the blue lines are rivers, and the black line is shoreline.



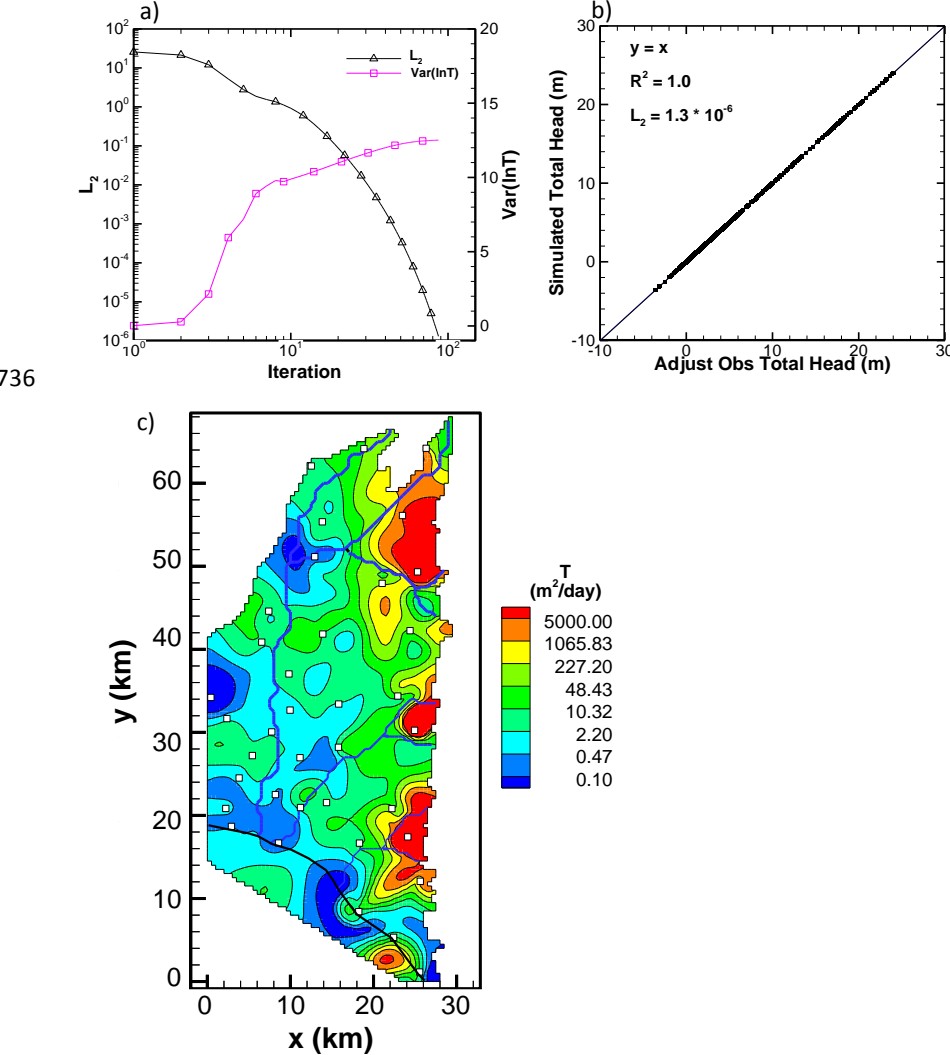



Figure 8. The estimated hydraulic transmissivity $T$ (m$^2$/day) field using observed head
in the field and new algorithm. a) The evolutions of mean squared error between the
observed and simulated heads ($L_2$ norm) and the spatial variance of ln$T$ (Var ln$T$)
during the calibration process. b) The calibrated heads verse adjusted observed heads
(observed head + estimated error) of the final iteration. c) The final estimated $T$ field.
The white squares represent wells, the blue lines are rivers, and the black line is
shoreline.




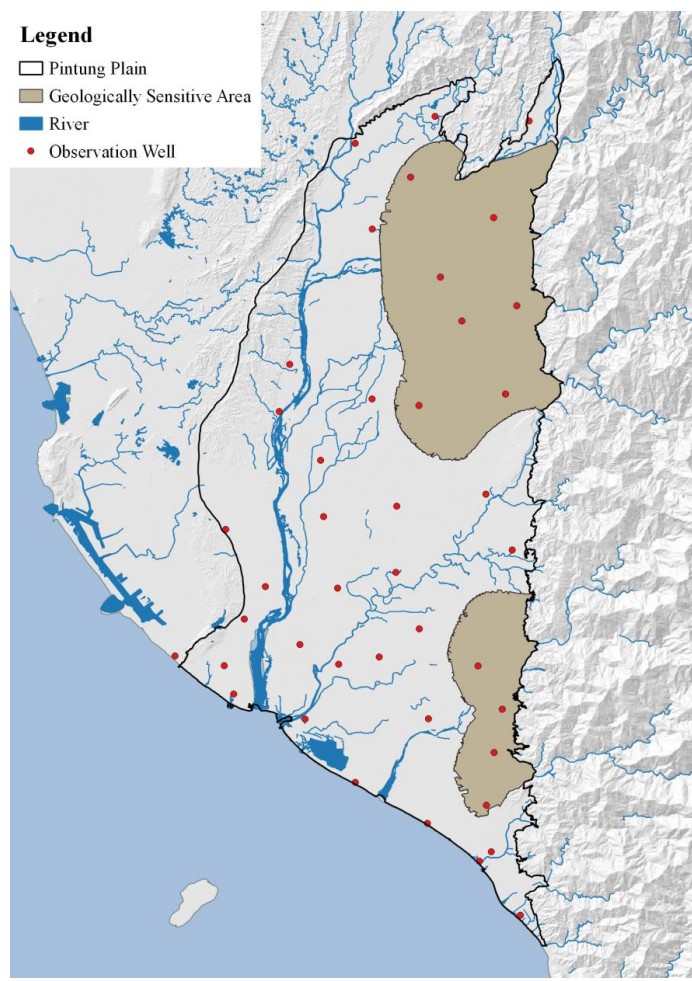

Figure 9. Geological sensitivity regions delineated by the Department of Central
Geology Survey, Taiwan.



**Code/Data availability**


The code and data are available upon the request through corresponding author.

**Author contribution**
Y.-L. Wang designed the study, carried out the analysis, interpreted the data, and
wrote the paper. T.-C. Jim Yeh and J.-P. Tsai provided the financial support and helped
finalize the paper.

**Competing interests**
The authors declare that they have no conflict of interest.