# Peer review of "A reduced-order model for dual state-parameter geostatistical inversion"

_Hydrology and Earth System Sciences, 2019_

## Referee Comment (RC1) · Anonymous Referee #1 · 12 Jan 2020

The abstract reads poor, and does not provide any reasonable scientific rationale. It uses unexplained terms, sich as "brick domain". I do not understand what is the problem at all about irregular domains. Also, it is not clarified what is a dual state-parameter inverse problem - do the authors refer to data assimilation? An abstract should raise research questions, but not questions about what the authors want to achieve. When even the summary of ideas is confuse, how can the paper contents be clear? Hence, I am afraid that this material is not ready for review yet.

---

## Author Comment (AC1) · 15 Jan 2020

Dear reviewer: Thanks for your comments. This study addressed the overfitting problem in geostatistical inversion approach. The researchers need to manually select the estimated parameters based on their experience to avoid over-calibration, an annoying issue when performing the ensemble algorithm or Monte Carlo simulation. That is to say, the previous methods are not fully automatic and objective. This study overcome this issue by proposing a reduced-order geostatistical inversion algorithm that consider input uncertainty. We further used two synthetic and one real case studies to examine the proposed method. The results of this study show that the proposed method indeed provide a more stable and objective estimate of the parameter fields than the approach without considering input uncertainty. Most importantly, the developed algorithm is to-

tally automatic and no labor requirement during the inversion procedure.

Regarding to the brick domain, here we refer to the rectangular domain in 2-D (or brick in 3-D). They are described in the introduction section. Previous reduced order algorithms require rectangular domain to efficiently decompose the unconditional co-variance matrix. This requirement comes from the derivation of analytic eigenvalue and eigenvector of a separable exponential function. The details are discussed in the introduction and methodology sections as well. The word "dual state-parameter" is not first used by this study. This word is frequently used in the previous studies related to parameter and state estimation (e.g., Moradkhani et al. 2005, Sorooshian et al. 2008, Lü et al. 2011, and Lü et al. 2013). Since this study address the state and parameter estimation in the field of subsurface characterization, we adopt this word "dual state-parameter" in this study without further explanation. However, according to your comment, we will further explain it in the introduction section in the next round of review process.

Moradkhani, H., Sorooshian, S., Gupta, H. V., & Houser, P. R. (2005). Dual state–parameter estimation of hydrological models using ensemble Kalman filter. Advances in water resources, 28(2), 135-147.

Sorooshian, S., Hsu, K. L., Coppola, E., Tomassetti, B., Verdecchia, M., & Visconti, G. (Eds.). (2008). Hydrological modelling and the water cycle: coupling the atmospheric and hydrological models (Vol. 63). Springer Science & Business Media. Lü, H., Yu, Z., Zhu, Y., Drake, S., Hao, Z., & Sudicky, E. A. (2011). Dual state-parameter estimation of root zone soil moisture by optimal parameter estimation and extended Kalman filter data assimilation. Advances in water resources, 34(3), 395-406. Lü, H., Hou, T., Horton, R., Zhu, Y., Chen, X., Jia, Y., ... & Fu, X. (2013). The streamflow estimation using the Xinanjiang rainfall runoff model and dual state-parameter estimation method. Journal of Hydrology, 480, 102-114.

[Figure]

622, 2020.

---

## Referee Comment (RC2) · Anonymous Referee #2 · 23 Jan 2020

Dear authors,

I read with interest your submission to HESS. The work can potentially make significant advances in solving inverse problems in hydrogeology and its topic is relevant to the readership of HESS. A robust inversion algorithm is developed to automatically account for data errors by reducing the number of parameters (and therefore avoid over-fitting) using SVD. The new approach is also efficient enough so that the derivation of adjoint equations no longer necessary for large highly parameterized inverse models.

The presentation of this work (esp. the abstract and conclusion) needs considerable improvement. There are too many obvious English errors and illogical sentences (see partial list below). I recommend having someone editing the paper before the next submission. I agree with RC1 that the abstract needs to be rewritten and have key

terms defined. Key findings such as computational savings should be quantified in the abstract. The conclusion section failed to highlight its major contribution (e.g. the main contribution of this paper is not to account for subsurface heterogeneity!). Lastly, perhaps some flowchart summarizing the method will be helpful.

I will be pleased to see this work published in HESS but some major revisions are needed to improve its presentation.

Overall assessment:

The methods section is difficult to follow. A better approach would be, without losing generality, introduce hydraulic tomography and SSLE first and then go into the SVD of error covariances.

In the results section, there is no mention of how many leading eigenvalues (or

The issue with brick/regular/rectangular domain mentioned in the abstract and L119-126 doesn't seem to be entirely justified. Is the matrix manipulation method shown here applied to hydrology problems the first time? Can they be readily applied to existing methods listed in the above lines? How useful is the matrix manipulation to eliminate cells compared with more pragmatic approaches such as using a larger domain and then set inactive cells in a flow problem?

Specific comments:

Please use the HESS template without any modification for resubmission.

Pay attention to these errors, e.g.: + "approach stable" or "approach steady" or "are steady" + "unknow" + "as the result" + "numeric errors" + "a bunch of realizations"

I can't follow these sentences: e.g. most of the abstract, L153, 222

Consider putting some lengthy derivation in the appendix

Title, L339: the entire article has no mention of dual state-parameter, revise

L40: what does "prefer scale" mean? I also find this paragraph quite ambiguous and not justifying high-resolution subsurface characterization

L57: computationally

L63: Afterwards

L67: give details

L70-71: Use Big O notation for computational cost

L73: by how much

L73 onwards:

L77: the finite-difference approach

L79: takes advantage

L92-96: it's not good enough to just list the existing methods. an assessment of their characteristics/strengths/ weaknesses is needed. A table may be helpful.

L105: in practice

L108: specifically

L110-L118: Do you think modelling the measurement errors (as in ERT literature) is a more straightforward way to solve the problem? also, check out "Robust inversion in ERT"

L142: Add a method overview

L197 onwards: needs to introduce the error formulation... consider using curly brackets under parts of the equation to introduce terms such as $W$ and $R_{hh}$

L280: Did you use the analytical solutions here?

L309: should specify which computation time refers to which regime

L311: Is it just a differencing scheme, why call it a perturbation approach?

L314 $g^{(r)}$ : confusing symbol, used $g$ as eigenvectors before. Use another symbol or font

L326-327, 335-336, section 2.5: This needs to be highlighted somehow! e.g. in a table and recapped in conclusions/abstracts

L328-336: Did you use this more accurate version here?

L340: how do you get this number?

L346: elegant?? do you mean the governing equation is linear or K does not depend on H?

L352: straightforward instead of easy

L411: in other words

L474: the increase in

L484: remove the before similar

L488: cite map – so that readers can know how the map is derived

L491: you mean deposits?

L493: you mean "collocated" instead of "in parallel"?

Fig 2 and 4: please mark which iteration is the used for the "best" iteration (in the caption or a vertical line in (a))

Fig 2 and 3, 4 and 5: Can't the modeller just saves all the outputs from the old algorithm and pick the best one?

Fig 8: R2=1.0 and y=X trendline– too good to be true? Please double-check or add digits

[Figure]

---

## Author Comment (AC2) · 25 Feb 2020

Dear authors,

I read with interest your submission to HESS. The work can potentially make significant advances in solving inverse problems in hydrogeology and its topic is relevant to the readership of HESS. A robust inversion algorithm is developed to automatically account for data errors by reducing the number of parameters (and therefore avoid over-fitting) using SVD. The new approach is also efficient enough so that the derivation of adjoint equations no longer necessary for large highly parameterized inverse models.

*Reply: We are glad to receive your constructive comments. They improve the organization and presentation of manuscript. We have revised the manuscript, especially the abstract and conclusion sections.*

The presentation of this work (esp. the abstract and conclusion) needs considerable improvement. There are too many obvious English errors and illogical sentences (see partial list below). I recommend having someone editing the paper before the next submission. I agree with RC1 that the abstract needs to be rewritten and have key terms defined.

*Reply: Thanks. We have revised the manuscript and had a native speaker editing the paper.*

Key findings such as computational savings should be quantified in the abstract.

*Reply: Thanks. They are included based on your specific suggestion for L326-327, 335-336, section 2.5. The revised abstract is:*

*"Geostatistical inverse methods usually permit numerous spatial correlated parameters to account for subsurface heterogeneity and predict state variability properly. However, a large number of unknown parameters (nf) causes considerable storage burden and computational complexity (e.g., evaluating the sensitivity efficiently) that challenge the application of classical inverse modeling techniques. In addition, the convergence criteria based on relative changes in parameter variance or fitness of observed and simulated states usually lead to under- or over-calibration. The best estimate, which is commonly selected manually, heavily relies on personal judgment. Accordingly, singular value decomposition (SVD) is employed, and the state error is estimated to reduce memory usage, improve computational efficiency, and stabilize the estimate. Specifically, the parameter covariance matrix is projected to the orthonormal basis (nr) through an analytical solution, and only the components that explain most of the original matrix structure are retained. The matrix size decreased from nf×nf to nf×nr. The covariance on the orthonormal basis is further evaluated directly through the finite difference scheme, and the sensitivity calculation is omitted. This approach*

*only requires performing the forward model run in the order of nr times. Finally, the parameters and errors of states are sequentially updated to leverage the over-fitting problem and accelerate the convergence by eliminating the unnecessary iteration. The computational advantages of the proposed reduced-order inverse algorithm are demonstrated through numerical and field case studies. Analysis suggests that the stability of model convergence dramatically improves. The estimated parameter values also stabilize to a reasonable order of magnitude. The memory requirement considerably diminishes, while the spatial resolution of the estimate is maintained. The proposed method benefits multi-discipline scientific problems and is especially useful and convenient for assimilating different types of measurements."*

The conclusion section failed to highlight its major contribution (e.g. the main contribution of this paper is not to account for subsurface heterogeneity!).

*Reply: Thanks. We have revised the conclusion based on your specific suggestion for L326-327, 335-336, section 2.5. The revised conclusion is:*

*"A reduced-order geostatistical model is developed in this study to leverage the stability of model convergence, storage burden, and computational complexity. The robustness of model convergence is considerably improved by estimating the state errors in the inverse modeling process. The final estimated parameter values stabilized to a reasonable order of magnitude. The memory burden (i.e., high dimensional parameter covariance) and the computational complexity of sensitivity are also relieved by approximating the parameter covariance matrix through a limited number of leading eigenvalues and eigenvectors. The matrix storage is reduced from nf×nf (number of unknown parameters) to nf×nsvd (number of retained eigens). The sensitivity is no longer evaluated and replaced by the direct evaluation of cross-covariance through the finite differencing method. The additional computational cost involved performing the forward model nsvd (number of stored eigens) × nevent (number of pumping or injection events) times. Finally, the proposed algorithm will be valuable and attractive for multi-discipline scientific problems due to the stable and robust convergence of the proposed method and the efficient evaluation of cross-covariance. Particularly, the proposed algorithm is beneficial for assimilating different types of state measurements."*

Lastly, perhaps some flowchart summarizing the method will be helpful.

*Reply: Thanks for your suggestion. A flowchart is included in the revised manuscript.*

[Figure]

I will be pleased to see this work published in HESS but some major revisions are needed to improve its presentation.

*Reply: Thanks for the recommendation. We have rewritten and reorganized the manuscript based on your suggestions and comments.*

Overall assessment:

The methods section is difficult to follow. A better approach would be, without losing generality, introduce hydraulic tomography and SSLE first and then go into the SVD of error covariances.

*Reply: Thanks. We have revised the method section and add a flowchart to compare the previous and proposed algorithm. The section is organized as SLE inverse algorithm, data error and its covariance, and covariance matrix reduction using SVD.*

In the results section, there is no mention of how many leading eigenvalues (or.

*Reply: Thanks. The leading eigenvalues used in both synthetic and field cases are 250. These eigenvalues and eigenvectors explain 84 % of original matrix structure. They are included in line.*

The issue with brick/regular/rectangular domain mentioned in the abstract and L119-126 doesn't seem to be entirely justified. Is the matrix manipulation method shown here applied to hydrology problems the first time? Can they be readily applied to existing methods listed in the above lines?

*Reply: Thanks for the suggestion. SVD is not a computational affordable method to evaluate the eigenvalue and eigenvector of covariance matrix utilized in the highly parameterized inverse problem (the number of unknowns may be more than ten thousand). Thus, we need to utilize the analytical solution for SVD decomposition. The analytical method is derived based on two assumptions. One is the brick grid and the other is the brick domain. We have revised the abstract and introduction to clarify this issue.*

How useful is the matrix manipulation to eliminate cells compared with more pragmatic approaches such as using a larger domain and then set inactive cells in a flow problem?

*Reply: They are mathematically identical. We just create a larger domain and remove the inactive cells out of the matrix.*

Please use the HESS template without any modification for resubmission.

*Reply: Thanks for your comment, we will use the HESS template in our revised manuscript.*

Pay attention to these errors, e.g.: + "approach stable" or "approach steady" or "are steady" + "unknow" + "as the result" + "numeric errors" + "a bunch of realizations"

*Reply: Thanks. We have corrected these errors in the revised manuscript.*

I can't follow these sentences: e.g. most of the abstract, L153, 222

*Reply: We have revised these sentences as the following:*

*Abstract: Please refer to the revised abstract above.*

*L153: The singular value decomposition (SVD) is employed to reduce the order of the parameter covariance, leading to less memory requirement and more computational efficiency in inverse exercise. In addition, the algorithm estimates the data error, which stabilized the estimated parameter field and improves the convergence stability.*

*L222: After the model structure (i.e., updated parameter field) improves, the uncertainty of simulated head ( $\mathbf{R}_{hh}^{(r)}$ ) reduces and the algorithm trusts the observation ( $\mathbf{h}^{*}$ ) less than the prediction ( $\mathbf{h}^{(r)}$ ).*

Consider putting some lengthy derivation in the appendix

*Reply: Thanks for your suggestion. We have reorganized the method section. Some of the derivations are moved to the appendix.*

Title, L339: the entire article has no mention of dual state-parameter, revise

*Reply: Thanks for your comments. We have removed dual state-parameter from the manuscript, and the title has been also revised.*

L40: what does "prefer scale" mean? I also find this paragraph quite ambiguous and not justifying high-resolution subsurface characterization

*Reply: We have modified the paragraph as following:" Understanding the spatial distribution of the site-specific hydrological parameters (e.g., hydraulic conductivity and specific storage) is one of the important foundations to design a successful management strategy."*

L92-96: it's not good enough to just list the existing methods. an assessment of their characteristics/strengths/ weaknesses is needed. A table may be helpful.

*Reply: Thanks for the suggestion. We have included the assessment and a table. The revised assessment is:*

*"Sun and Yeh (1990) employed the adjoint approach to evaluate sensitivity. This approach reduces the cost of running the forward model to less than nd times. Liu et al.*

*(2014), Kitanidis and Lee (2014), Li et al. (2015), and Zha et al. (2018) projected the parameter covariance matrix on the orthonormal basis and directly evaluated the covariance between parameter and state. The computational cost of running the forward model is reduced to the order of the number of basis (nr) by using the adjoint (Liu et al., 2014) and finite differencing (Kitanidis and Lee, 2014; Li et al., 2015; and Zha et al., 2018) approach.*

*Besides, Saibaba and Kitanidis (2012) used the hierarchical nature of matrices to accelerate the computation of dense matrix–vector products. Similarly, Li et al. (2014) used the hierarchical matrix technique to rewrite the Kalman filtering equations into a computationally efficient counterpart, and Ghorbanidehno et al. (2015) extended the hierarchical matrix technique to the general case of non-linear dynamic systems. The hierarchical technique reduces the computational complexity from $O(nf{\times}nf{\times}nd)$ to $O(nf{\times}nd{\times}lognf)$ when constructing the covariance matrix between parameter and state.*

*In addition, many approaches relieve the memory requirement by reformulating the covariance matrix. For example, Nowak and Litvinenko (2013) combined low-rank approximations to the covariance matrices with fast Fourier transform; Kitanidis (2015) decomposed the covariance matrix through some orthonormal basis and suggested that the choice of basis can be tailored to the problem of interest to improve the estimation accuracy; Li et al. (2015) used discrete cosine transform to compress the model covariance matrix; Zha et al. (2018) employed the Karhunen–Loeve expansion and Kitanidis and Lee (2014) applied low-rank factorization to compress the parameter covariance matrix. These methods reduce the storage cost from $nf{\times}nf$ to $nf{\times}nr$. Zunino and Mosegaard (2019) utilized the Kronecker product decomposition to reduce the storage of cross-covariance between state and parameter.*

*Performing the inverse operation efficiently is problematic when the number of state nd is large. Saibaba and Kitanidis (2012) and Liu et al. (2014) used a matrix-free Krylov subspace approach (e.g., restarted GMRES[50] and MINRES) to reduce the storage and computational cost. Specifically, this method uses a predefined low-rank representation of the prior covariance matrix to establish a close approximation of the inverse state covariance matrix and eliminates the storage of state covariance matrix. However, additional forward model runs are required. A randomized algorithm is another tool used to reduce the size of the state covariance matrix (Lin et al., 2017). This algorithm projects the state and governing equations to several subspaces through a sketch matrix. The size of the state covariance matrix reduces to $nr{\times}nr$.*

*In addition to reformulating the covariance matrix, several methods reduce the computational complexity of the governing equation. For instance, the temporal moment eliminates the temporal derivative term in the governing equation (Cirpka and Kitanidis, 2000; Nowak and Cirpka, 2006; Yin and Illman, 2009). Galerkin projection*

*reduces the matrix size (i.e., A and B matrices) of a sparse linear system equation AX = B by projecting the state variable to the orthonormal basis (Liu et al., 2013). Lin et al. (2016) also projected the sparse linear system equation to the Krylov subspace with the Golub–Kahan–Lanczos bidiagonalization technique and reduced its matrix size. The combination of random mixing with the Whittaker–Shannon interpolation also reduces the computational cost of forward modeling (Horning et al., 2019).*

*The storage and computational complexity of the aforementioned methods are summarized in Table 1."*

Table 1.

| Method | Storage | Comp. Cost | |
|---|---|---|---|
| **Parameter/Model Covariance Matrix** $\varepsilon_{ff}$ | | | |
| Full matrix | $n_f \times n_f$ | - | |
| Any orthonormal basis | $n_f \times n_r$ | - | Kitanidis (2015) |
| Low rank factorization | $n_f \times n_r$ | - | Kitanidis and Lee (2014) |
| Discrete cosine transform | $n_f \times n_r + n_r$ | - | Li et al. (2015) |
| Karhunen-Loeve expansion | $n_f \times n_r + n_r$ | - | Zha et al. (2018) |
| **State Covariance Matrix** $\mathbf{R}_{hh}$ | | | |
| Full matrix | $n_d \times n_d$ | - | |
| Randomize algorithm | $n_r \times n_r$ | - | Lin et al. (2017) |
| **Sensitivity** $\mathbf{J}_{fh}$ **and Covariance** $\mathbf{J}_{fh}^T \varepsilon_{ff}$ | | | |
| Finite difference | $n_f \times n_f$ | $n_f + 1$ | |
| Adjoint (linear diff. eq.) | $n_f \times n_d$ | $n_w$ | Sun and Yeh (1990) |
| Adjoint (nonlinear diff eq.) | $n_f \times n_d$ | $n_d$ | Sun and Yeh (1990) |
| Adjoint on orthonormal basis | $n_d \times n_r$ | $n_r$ | Liu et al. (2014) |
| Finite difference on orthonormal basis | $n_d \times n_r$ | $n_r \times n_{event}$ | Kitanidis and Lee (2014) Li et al. (2015) Zha et al. (2018) |
| **Forward Modeling** | | | |
| Galerkin projection | - | Size of AX=B | Liu et al. (2013) |

| | Golub-Kahan-Lanczos bidiagonalization | - | Size of AX=B | Lin et al. (2016) |
|---|---|---|---|---|
| | Temporal moment | - | Steady state | Cirpka and Kitanidis (2000) Nowak and Cirpka (2006) Yin and Illman (2009) |
| **Matrix Multiplication** | | | | |
| | $\mathbf{J}_{fh}{}^{T}\mathbf{\varepsilon}_{ff}$ | - | $O(n_f \times n_f \times n_d)$ | |
| | Hierarchical matrix | - | $O(n_f \times n_d \times \log n_f)$ | Saibaba and Kitanidis (2012) Li et al. (2014) Ghorbanidehno et al. (2015) |
| **Inverse Matrix** $\mathbf{R}_{hh}^{-1}$ | | | | |
| | Cholesky decomposition | $n_d \times n_d$ | $O(0.33\, n_d^3)$ | |
| | QR decomposition | $n_d \times n_d$ | $O(1.33\, n_d^3)$ | |
| | Krylov subspace (MINRES, GMRES50) | 0 | $2n_s + n_r + 3$ | Saibaba and Kitanidis (2012) Liu et al. (2014) |

1. $n_r$ is the reduced dimension. It could be the number of frequencies (Li et al., 2015), orthonormal basis (Liu et al., 2014; Kitanidis 2015; Zha et al., 2018), ranks (Saibaba and Kitanidis, 2012; Kitanidis and Lee, 2014), or the size of sketch matrix (Lin et al., 2017).

2. Comp. Cost represents the computational cost. It could be either number of forward model runs or computational complexity.

3. $n_s$ is the number of iterations in MINRES or GMRES50. $n_f$ is the number of unknown parameters. $n_d$ is the number of states. $n_w$ is the number of wells or unique measurement locations. $n_{event}$ is the number of pumping test.

L110-L118: Do you think modelling the measurement errors (as in ERT literature) is a more straightforward way to solve the problem? also, check out "Robust inversion in ERT"

*Reply: We can remove the outlier from dataset or take more measurement to eliminate the effects of outlier. However, as demonstrated in the measurement-error free synthetic example, the estimate is still over-calibrated. We have checked some papers related to the robust inversion in ERT. These papers gradually eliminated the outlier of the observation data and did not directly consider the state error.*

*Morelli, G., & LaBrecque, D. J. (1996). Advances in ERT inverse modelling. European Journal of Environmental and Engineering Geophysics, 1(2), 171-186.*

*Morelli, G., & LaBrecque, D. J. (1996, April). Robust scheme for ERT inverse modeling. In 9th EEGS Symposium on the Application of Geophysics to Engineering and Environmental Problems (pp. cp-205). European Association of Geoscientists & Engineers.*

L142: Add a method overview

*Reply: Thanks. The overview is added. "In this study, we adopt the subsurface characterization technique (successive linear estimator, SLE) developed by Yeh et al. (1996). This algorithm conceptualizes the hydraulic parameters and state variables as the spatial stochastic process and assumed that they can be characterized by some statistical information (e.g., mean, variance, and spatial correlation function). Thereafter, we improve the algorithm stability by modeling the measurement error. We further reduce the computation complexity (e.g., sensitivity) and storage burden by singular value decomposition (SVD). A flowchart summarizing the differences between the old and new algorithms is presented in figure 1."*

L197 onwards: needs to introduce the error formulation... consider using curly brackets under parts of the equation to introduce terms such as W and Rhh

*Reply: The error formulation is introduced in the required input section (section 2.3).*

L280: Did you use the analytical solutions here?

*Reply: Yes, we use the analytical solution for SVD decomposition.*

L309: should specify which computation time refers to which regime

*Reply: Thanks for your comment, and they are included in the revised manuscript. "For instance, if the parameters are independent with state variable (e.g., saturated flow equation in a confined aquifer) and the boundary conditions are identical between different pumping tests, only nw times of adjoint forward model is required. If the boundary conditions vary between different pumping tests, we need to run the forward model nw × nevent (number of pumping or injection events) times. If the parameters dependent on the state variable (e.g., unsaturated flow equation or flow through an unconfined aquifer), nd times of forward modeling are necessary."*

L311: Is it just a differencing scheme, why call it a perturbation approach?

*Reply: Thanks for your comment. We have modified it to finite differencing scheme.*

L314 g(r) : confusing symbol, used g as eigenvectors before. Use another symbol or Font

*Reply: g refers to the eigenvector. We evaluate the covariance on the orthonormal basis and skip the sensitivity evaluation.*

L326-327, 335-336, section 2.5: This needs to be highlighted somehow! e.g. in a table and recapped in conclusions/abstracts

*Reply: They are included in the abstract and conclusion.*

*Abstract: The covariance on the orthonormal basis is further evaluated directly through the finite difference scheme, and the sensitivity calculation is omitted. This approach only requires performing the forward model run in the order of nr times.*

*Conclusion: The sensitivity is no longer evaluated and replaced by the direct evaluation of cross-covariance through the finite differencing method. The additional computational cost involved performing the forward model nsvd (number of stored eigens) × nevent (number of pumping or injection events) times.*

L328-336: Did you use this more accurate version here?

*Reply: Yes, we have included the following description. "Although using the adjoint approach is more computational efficient in our example (i.e., linear diffusion equation), for demonstration purpose, we use the central difference scheme to evaluate the squared root of cross-covariance $\mathbf{H}_{fh}^{(r)}$."*

L340: how do you get this number?

*Reply: We have included the following citation: "S. Lahabar and P. J. Narayanan, "Singular value decomposition on GPU using CUDA," 2009 IEEE International Symposium on Parallel & Distributed Processing, Rome, 2009, pp. 1-10."*

L346: elegant?? do you mean the governing equation is linear or K does not depend on H?

*Reply: We have changed the elegant to linear.*

L352: straightforward instead of easy

*Reply: We have corrected it.*

L488: cite map – so that readers can know how the map is derived

*Reply: The map is derived from the following paper: Tsai, J. P., Chen, Y. W., Chang, L. C., Chen, W. F., Chiang, C. J., Chen, Y. C.: The assessment of high recharge areas*

*using DO indicators and recharge potential analysis: a case study of Taiwan's Pingtung plain. Stoch. Env. Res. Risk A., 29(3), 815-832, 2015.*

L491: you mean deposits?
*Reply: We have modified it.*

L493: you mean "collocated" instead of "in parallel"?
*Reply: We have modified it.*

Fig 2 and 4: please mark which iteration is the used for the "best" iteration (in the caption or a vertical line in (a))
*Reply: We have modified it.*

Fig 2 and 3, 4 and 5: Can't the modeller just saves all the outputs from the old algorithm and pick the best one?
*Reply: Fig 3 and 5: These figures are from the new algorithm. The final iteration is the best estimate field. No need to manually pick the best one.*
*Fig 2 and 4: Please be aware that without knowing the true field, it is impossible to select the best estimate. One can only subjectively choose a reasonable estimate based on the personal judgement.*

Fig 8: R2=1.0 and y=X trendline– too good to be true? Please double-check or add digits
*Reply: It is the correct trendline.*